# Safety Issues in Buckling of Steel Structures by Improving Accuracy of Historical Methods

**DOI:** 10.3390/ijerph182212253

**Published:** 2021-11-22

**Authors:** Juan Carlos Pomares, Javier Pereiro-Barceló, Antonio González, Rafael Aguilar

**Affiliations:** 1Civil Engineering Department, University of Alicante, Carretera San Vicente del Raspeig s/n, 03690 San Vicente del Raspeig, Spain; javier.pereiro@ua.es (J.P.-B.); antonio.gonzalez@ua.es (A.G.); 2Engineering Department, Pontificia Universidad Católica del Perú, San Miguel, Lima 15088, Peru; raguilar@pucp.edu.pe

**Keywords:** safety, buckling, historical, column, steel, collapse

## Abstract

Buckling of structural elements is a phenomenon that has great consequences on the bearing capacity of structures. Historically, there have been serious buckling-related structural accidents that have resulted in loss of human lives and high material costs. In this article, an attempt is made to perform a historical analysis of the diverse models that experts have been using in designing and calculating compression buckling of simple metallic elements in the last 275 years. The analysis covers the lapse from the mid-18th century, in which the pioneers in this classic field of structural design are located, up to the present, highlighting the main standards that have been applied to steel structural analysis in the past and at present all over the world. What the study tries to provide is an overall view and a sense of continuity of the methods used for improving structural safety regarding buckling failures in the last three centuries. Each analyzed buckling model is compared with the results of a numerical finite element model of compressed steel columns. Finally, the conclusion reached is that in the last one hundred years, the convergence of solutions proposed in the field is gradually greater and more accurate.

## 1. Introduction

Throughout history, numerous accidents and collapses of structures have occurred due to buckling of steel structural elements, which resulted in loss of human lives and high material costs. Several bridge accidents, such as Dee Bridge, 1847, Tay Bridge, 1879, Quebec Bridge, 1907, and Tacoma Bridge, 1940, stimulated a considerable amount of research into the buckling behavior of steel members [1]. Augenti and Parisi [2] discussed the collapse of a long-span steel roof structure which fell down suddenly during construction as a result of an out-of-plane buckling phenomenon induced by wind. Al-Marwaee [3] stated that the World Trade Centre towers were designed with steel columns braced against buckling by the floors. The high temperatures in the fires weakened the steel brace systems and columns, causing the buckling length of columns to increase and, consequently, the onset of buckling.

The first historical, scientifically valid reference to the concept of maximum axial load capable of supporting a column subjected to simple compression comes from the hands of Leonardo da Vinci, the Renaissance genius, who, in the second half of the 15th century, deals with the strength of the columns in the following terms: “This varies inversely proportional to its length, but directly proportional to its cross section” [4]. If Leonardo’s words were transformed into today’s mathematical language, they would be
(1)$$P_{crit}=\frac{k\cdot A}{L}$$
where
$P_{crit}$Maximum critical load which depletes the column.kUndetermined coefficient, with units of force over length.ACross-sectional area of the column.LLength of the column.

Later in 1638, the great scientist Galileo Galilei, in his book *Dialogues Concerning Two New Sciences* [5], posed several problems that nowadays would pertain to the field of materials science; among them, that which enquired about the maximum axil load needed to deplete a column subjected to simple compression. In Proposition 1, Galileo, through his characters, wonders about the axil load that would support a beam, and reaches the conclusion that the load decreases as the beam’s geometrical slenderness increases. However, he did not ground it mathematically. Galileo, as well as Leonardo, left this problem unsolved. Galileo also presented similar problems, such as that of the cantilever, in another seven new propositions. Galileo himself did provide some type of solutions to such problems, but none stood the passage of time.

In 1729, the Dutchman Petrus van Musschenbroek [6] (PvM) of the University of Utrecht published a voluminous book containing dissertations on several different topics. Of special interest is the section entitled: “lntroductio ad cohaerentiam corporum firmorum”, in which PvM reported experiments carried out mainly on pieces of wood, which he subjected to tensile, bending, and compressive stresses. By testing pieces of wood of different dimensions in compression, PvM identified that, during the loading process and before reaching breakage, the columns underwent lateral bending and postulated a general law derived from the experiments, according to which the resistance of a column is inversely proportional to the square of its length.

A century passed by, and in 1744 [7] the great Swiss mathematician Leonhard Euler tackled the problem of the column model with absolute mathematical precision. He proposed his celebrated buckling critical load equation for an elastic cantilever column subjected to centered compression, which, using the original terminology, appears in the following equation:(2)$$P_{crit}=\frac{C\cdot \pi^{2}}{4\cdot \;L^{2}}$$
where
*C* Material dependent coefficient and cross section, in units of force multiplied by length squared.

Euler later narrowed the coefficient “*C*” down to the product of *E·k*^2^, “*E*” being a property of the strength of the material, and “*k*^2^”, a dimensional property of the column’s cross section.

For Euler’s coefficient, *E·k*^2^, to become today’s “*E·I*”, where “*E*” is the material’s modulus of longitudinal elasticity, the Young’s modulus, and “*I*” is the moment of inertia with respect to the axis buckling, this took place after a number of decades passed by. Meanwhile, the alternative was to apply Hooke’s law in combination with the correct assessment of the internal distribution of the loads in the bent member. Bernoulli and Coulomb made great contributions for the problem to be solved, but it was not until 1770–1773 that the great French mathematician Lagrange came up with a solution to the equation of the buckling critical mass in a biarticulated elastic column subjected to centered compression, which in today’s words would look similar to the following equation:(3)$$P_{crit}=\frac{\pi^{2}\cdot E\cdot I}{L_{k}^{2}}$$
where
 *E* Modulus of longitudinal elasticity of the column’s material. The Young’s Modulus. *I* Moment of inertia with respect to the axis buckling takes place. *L_k_* Buckling length of the column. In the case of pinned–pinned bar, it corresponds to the geometrical length, precisely because it is a pinned–pinned column.

In the study of a compressive buckling bar, Euler was the first to come up with the idea that the resistance of the column decreases with the square of its length, precisely because length is in the denominator. Da Vinci and Galileo suspected that resistance decreased with geometrical slenderness of the member, i.e., length to the first power.

Equation (3) can readily be transformed into critical compressive buckling stress by dividing both terms of the equation by the cross-sectional area and making some straightforward mathematical operations. It can be expressed as
(4)$$\sigma_{crit}=\frac{\pi^{2}\cdot E}{\lambda_{k}^{2}}$$
where the referents are the same as those for Equation (3): *σ_crit_* Maximum critical compression stress that triggers the buckling of the column. *λ_k_* Mechanical slenderness of the column.

The mechanical slenderness (*λ_k_*) for a linear member can be defined as the ratio between the effective length of the buckled column and the radius of gyration with respect to the main axis of inertia of the section under consideration. Moreover, the radius of gyration is, in turn, the square root of the ratio between the inertia of the section and its area. In numerical form, it can be expressed as Equation (5), where (“*β*”) is a nondimensional coefficient of buckling, assuming the member’s support conditions, 1 being the value for the pinned–pinned bar, and 2 for Euler’s cantilever bar.
(5)$$\lambda_{k}=\frac{L_{k}}{i}=\frac{\beta \cdot L}{\sqrt{\frac{I}{A}}}$$

Through their equations, Leonhard Euler and his successors introduced a series of concepts that, from that time to the present, have played a decisive role in the study of structural mechanics. They include “moment of inertia”, “radius of gyration of a flat section”, and “mechanical or geometrical slenderness of a straight guideline bar”. Surprisingly enough, Euler’s equation is one of the few that has stood unmodified for almost three centuries, and still applies.

As a curiosity, let us mention that, paradoxically, in Euler’s equation there is no room for the material’s compressive stress. For that reason, for slight slenderness, huge values are obtained. It took almost one hundred years to understand that Euler’s equation is only applicable to steel columns whose mechanical slenderness surpassed the slenderness limit of 100.

In 1826, L. Navier (1785–1836) pointed out the fact that the values of the critical force according to Euler form the upper limit of the resistance of real columns, due to the assumption of ideal elasticity of the material.

The concept of mechanical slenderness (*λ_k_*), a key factor in buckling, was first defined in 1845 by E. Lamarle [4].

## 2. Development of Steel Construction: 19th, and Beginnings of the 20th, Century

With the advent of iron as a new structural material for construction both in civil engineering and in building construction in the middle of the 19th century in Europe, a new series of structural typologies where metallic members acquire particular relevance appear as a result.

These new metallic members that initially were made of cast iron were much thinner compared to previous members, basically made of stone as were the case of the pillars and columns made of marble, brick, wood, or masonry using some other material.

The slenderness of the new cast iron elements carried with them the idea of buckling for loads that were less than that they could support in simple compressive stress of the material of which they were made: cast iron in the first place, and later, steel.

In previous buildings where the slenderness of walls, pillars, and columns rarely surpassed a geometric ratio of 1:10, i.e., a geometrical slenderness less than 10—which would equate a mechanical slenderness in the order of 35 for a rectangular section whose total width was at its greatest 10 times its thickness—the problem of compressive buckling of any of the members was seldom seen as a real problem. Should any designer any time dare to surpass the recommended slenderness, the trial methods would tell him where the limits were to be contemplated in his designs.

One of the first bibliographic references to more complex equations or formulae that took buckling in cast iron columns into consideration in Spain came thanks to José Marvá y Mayer, Engineer General [8], who, around the year 1910, makes some reference, mainly taken from North American, British, and French research, to formulae for the dimensions of cast iron columns.

In his reference work [8], José Marvá provides a number of formulae to be applied to find out the dimensions of solid or hollow circular cast iron columns using expressions in line with the following equation. He calls it Hodgkinson’s formula:(6)$$P_{o}=k\cdot \frac{d^{3.6}}{L^{1.7}}$$
where
 *P_o_* Fracture/buckling load, in kp. *d* Outer diameter of the column, in cm. *L* Length of the column, in dm. *K* Coefficient of values 10,320 for a fixed bar, 5954 for fixed–pinned, and 2977 for pinned–pinned.

The numerical coefficient “*K*” of the equation is dimensional. Marvá recommended a working design on the order of six times less than that of fracture. This formula worked for those columns whose length contained 25 and 60 times their diameter, precisely in the part of the column in which buckling was clearly appreciated, the reason being that the corresponding mechanical slenderness to such values would be located between 90 and 200, approximately.

Eaton Hodgkinson (1789–1861) [4] was a British self-taught engineer who brought forward a number of experimental formulae he obtained from his own tests. The formulae, widely used all through the 19th century, dealt with compressive buckling of such members as metallic columns.

In his work [8], Marvá refers to formulae proposed by the French researchers M. Barré and G.H. Lowe (1859) to apply to problems related to establishing the dimensions of wooden and steel columns.

G.H. Lowe proposes the following expression—the original wording of which remains to this day—for solid cast iron columns:(7)$$P_{0}=\alpha \cdot \frac{R^{\prime }\cdot w}{1.45+0.00337\cdot {\left[\frac{L}{d}\right]}^{2}}$$
where
 *P_0_* Maximum fracture load which the column can withstand. *α* Coefficient that depends on the boundary conditions, 1 for a fixed member, 4/7 for a fixed–pinned member, and 2/7 for a pinned–pinned member. *R′* Simple compressive fracture stress of the material the column is made of. *w* Cross-sectional area. *L* Member length. *d* Outer diameter of the column.

The geometrical slenderness squared in Equation (7) appears in the denominator of the previous expression to act as a corrector to the buckling stress.

Should the expression be termed in today’s terms, it would be as follows:(8)$$N_{b,Rd}=\alpha \cdot \frac{f_{yd}\cdot A}{1.45+0.00337\cdot {\left[\lambda_{g}\right]}^{2}}\;$$

Later, in the same work, for more general cases, Marvá [9] proposes a formula from Barré, which in the original terminology is
(9)$$P_{o}=\frac{R^{\prime }\cdot w}{1+0.000156\cdot \frac{L^{2}\cdot w}{I}}$$

What these Expressions (8) and (9) mean is the same as previous formulae. If this last equation were to be expressed in today’s terminology, it would be
(10)$$N_{b,Rd}=\frac{f_{yd}\cdot A}{1+0.000156\cdot \lambda_{k}^{2}}$$

Expression (10), which came about at the end of the 19th century, resembles today’s standards and more recent models. Obviously, a nondimensional coefficient for buckling “*χ*” less than the unit—the ratio between the critical load and the plastic load—has been introduced indirectly with the following value:(11)$$\chi =\frac{1}{1+0.000156\cdot \lambda_{k}^{2}}\leq 1$$

The coefficient “*χ*” is inversely proportional to the square of the mechanical slenderness. Consequently, as the critical axle decreases, the square of the mechanical slenderness increases.

In the second half of the 19th century and the first half of the 20th, researchers continued studying the phenomenon of buckling of actual metallic members, probably motivated by the great development metallic construction undergoing and its wide application to every structure in the fields of civil engineering and building construction.

In the 1860s, another researcher, William John Macquom Rankine, a Scotsman, made another important contribution for the understanding of the phenomenon of buckling of slender steel columns in his book *A Manual of Civil Engineering* [9]. The first edition of the very many it had dates back to 1862. Rankine refers to Lewis Gordon, who in turn took Eaton Hodgkinson’s experimental results as the basis of his work. Rankine proposed that when a column exhibits a mechanical slenderness of less than 105, the following formula, already converted to today’s terminology, should be applied:(12)$$P_{crit}=\frac{\;f_{yd}\cdot A}{1+c\cdot \lambda_{k}^{2}}$$
where “*c*” is a nondimensional coefficient that can have the following values:

*c* = 1 × 10^−4^, for steel.

*c* = 2 × 10^−4^, for cast iron and wood.

When the mechanical slenderness is greater than 105, Rankine suggests that Euler’s equation should be applied directly.

Equation (12), proposed by Rankine, is half-experimental and has a vague resemblance to Lowe and Barré’s previous proposals. Rankine’s expression in its various formulations was profusely used for the design of steel columns up to the first half of the 20th century, so much so that in the year 1929, the “American Institute of Steel Construction” [10] produced the following equation, which in tensile stress units (kp/cm^2^) and in today’s terms would be expressed as follows:(13)$$f=\frac{P_{crit}}{A}=\frac{1265}{1+\frac{L^{2}}{18,000\cdot r^{2}}}$$
where “*r*” is the radius of gyration smaller than the cross section of the steel column. In this case, the coefficient “*c*” of Rankine’s equation was 1/18,000 = 0.000056 with a yield stress of steel of 1265 kp/cm^2^ (126.5 N/mm^2^). It was not to surpass the value of 1050 kp/cm^2^ (105 N/mm^2^) for tensile working stress “*f*”, and the mechanical slenderness was to be less than or equal to 120.

Towards the end of the 19th century, very simple formulae were proposed with parabolic expressions for the buckling design of steel column, as was A. Ostenfield’s [11] in the year 1898, which stated that for ordinary steel, the equation looked similar to the following:(14)$$\sigma_{crit}=2650-0.09\cdot {\left(\frac{L}{k_{z}}\right)}^{2}$$
where
 *L* Member length. *k_z_* Radius of gyration of the section with respect to the buckling axis.

Equation (14) is expressed in the stress units of kp/cm^2^.

In the year 1886, the Slovak researcher Ludwig von Tetmajer (1850–1905) [12], having gone through a deep study of a great many experimental results, proposed a diagram for the critical buckling of steel, the most widely applied at the time in Germany and middle Europe, the A-37, whose yield stress was in the order of 240 MPa (2400 kp/cm^2^). When the members presented mechanical slenderness greater than 105, Euler’s equation was to be applied directly, since Euler’s hyperbola application limit is defined as
(15)$$\lambda_{lim}=\frac{L_{k}}{i}=\sqrt{\frac{\pi^{2}\cdot E}{f_{y}}}=\sqrt{\frac{\pi^{2}\cdot 210,000}{200}}\approx 105$$

When the members are characterized by a mechanical slenderness that falls between 0 and 105, the straight line of Tetmajer was proposed (Figure 1) by regression of his experimental data, which, in stress units of kp and cm squared, is the following:(16)$$\sigma_{crit}=\left[3100-11.4\cdot \frac{L}{i}\right]$$

Equation (16) is known as Euler–Tetmajer. Tetmajer also proposed a more elaborated parabolic expression to account for this area of buckling.

Around this time, in 1889, the researchers Considère and Engesser advanced separately that in Euler’s Equation (4), and for slenderness less than 100, the Young’s modulus to be used was to be less than the actual one, which they termed as effective modulus “*E_eff_*” and tangent modulus “*E_t_*”, respectively, to predict the behavior of the compressed member in the nonelastic zone, where Euler’s equation cannot apply.

About two decades later (1910), von Karman suggested his well-known double-modulus *E_R_* theory founded on Engesser’s theory and Considere’s idea [13].

Southwell (1914) introduced the concept of initial deformations in straight pieces under compression, in order to infer a first Euler load on the column [14].

Sometime later, in 1928, German standards for metallic construction stated that for steel type A-37 of yield stress 2400 Kp/cm^2^, the stress–mechanical slenderness diagram in Figure 1, which happens to be Tetmajer’s with some modification, was to be applied. The precise values of the mechanical slenderness in the points B and C of the diagram are exactly 61.4 and 104, respectively. The German legislation reporters rounded off to 60 and 100, with sound criteria. With Tetmajer’s contributions, the design and calculation of slender steel columns subjected to centered compression advanced by leaps and bounds.

To bring to an end this historical overview on equations for the design of slender steel column buckling, Timoshenko’s secant method formula needs to be noted, deduced forthright from his *Materials Resistance* [11]. The secant formula in today’s terms would be as follows:(17)$$\sigma_{max}=\sigma_{c}\cdot \left[1+\frac{e\cdot c}{r^{2}}\cdot sec\left(\frac{L}{2\cdot r}\cdot \sqrt{\frac{\sigma_{c}}{E}}\right)\right]$$
where the meanings of which are
 *σ_max_* Steel yield stress (*f_y_*). *σ_c_* Maximum compression stress which depletes the column. *e* Load eccentricity caused by the flexural momentum acting on the section, measured in relation to the section’s center of gravity. *c* Section’s center of gravity distance with respect to the most compressed fiber, for symmetric sections *c* = h/2 (half the section’s edge). *E* Longitudinal elasticity modulus of the column’s material. The Young’s Modulus. *r* Radius of gyration of the section with respect to the buckling axis. *L* Buckling length of the member.

The secant method formula has basically two shortcomings when applied to a slender steel column subjected to central compression. The first problem is that the equation is transcendent, i.e., the value of “*σ_c_*” must be obtained by successive numerical approximations. The second is that in cases when there is no exterior flexural moment applied to the section, there will be no eccentricity: “*e* = 0”, and so the formula cannot work. In such circumstance, Euler’s equation must be used straightforwardly. A possible solution to this problem—and only for columns subjected to central compression—could be that eccentricity equaled the initial geometric imperfection of the column, either “*e* = *L*/500 or *e* = *L*/300”, depending on the manufacturing and assembly tolerances. Finally, the type of section must be defined, as the secant method formula requires that the parameter’s value “*c*” be defined.

In the case of plate buckling, the earliest attempts were made by Bleich (1924), [15] who suggested the replacement of Young’s modulus in the elastic critical buckling stress of plates by a reduced modulus such as the tangent modulus or the secant modulus.

Using the Engesser–von Karman methods for the inelastic buckling of columns, and with reference to the buckling of stocky plates beyond the proportional limit, Lundquist (1939) was the first who derived the buckling equation for plates [16].

The axially loaded steel I-section column has a tendency to buckle by twisting on its own axis. In the case of thin-wall sections of certain cross section and length, twisting buckling may occur at lower loads than the bending or Euler buckling. This type of buckling was studied by Kappus (1938) [17].

Based on Wagner and Kappus theories, Goodier, in 1941 [18], investigated the behavior of columns which are torsionally weak. In 1946, Shanley stated that the tangent modulus *E_t_* is the correct effective modulus to be employed for buckling beyond the proportional limit and that the unloading of one side of the column does not occur until the tangent modulus load is reached [19,20].

## 3. The First Construction Standards, the Second Half of the 20th Century

It was in the year 1952 that the German standard DIN 4114 [21] introduced the concepts of buckling coefficient “ω” in the form of a table for the various types of steel existing at the time: the A-37 and the A-52. It also defined the concept of length of buckling in terms that practically have not changed today. The concept of buckling coefficients, “*ω*”, however, had already been proposed by the French researcher J. Dutheil [22] sometime around the year 1947.

In 1952 the German standards criteria were introduced in Spain, and then in the year 1962, the Spanish EM-62 [23] came to life, which, in its 1969 version, proposes the very same concepts for the buckling coefficient “*ω*” and for the length of buckling “*L_k_*”.

In the year 1972, the standard MV-103 [24] was published in Spain. This standard broadens and improves the previous EM-62 in issues related to the design and calculation of buckling in simple steel columns subjected to centered compression—which happens to be the same as that named NBE-EA-95 [25] that appeared in 1995. This standard, MV-103, stood the passage of time well into the year 2006, when it was superseded by the CTE-DB SE A (Steel) [26].

Both Spanish standards, the MV-103 and the NBE-EA-95, introduced nondimensional coefficients for buckling “ω”; coefficients greater than the unit, arranged in three tables for the three types of steel currently in existence in Spain: A-37, A-42, and A-52, each of which show a yield stress of 2400, 2600, and 3600 Kp/cm^2^, respectively.

In standards MV-103 and NBE-EA-95, the buckling test is carried out by satisfying the following inequality:(18)$$\frac{P^{*}\cdot \omega }{A}\leq \sigma_{adm}$$
where
 *P** Maximum design load the column can withstand. *ω* Nondimensional buckling coefficient greater than or equal to the unit, a function of the type of steel, yield stress, and mechanical slenderness of the member. *A* Gross cross-sectional area. *σ_adm_* Tolerable steel strength equal to its yield stress.

As shown in Equation (5), the mechanical slenderness is a function of the buckling length and the radius of gyration with respect to the buckling axis or buckling plane. The buckling coefficients “*ω*” of previous Spanish standards, MV-103 and NBE-EA-95, can, not only readily but also exactly, be obtained through numerical calculations rather than through the proposed tables in the standards, applying the following expressions:(19)$$a=\frac{\sigma_{adm\cdot }\;\lambda_{k}^{2}}{\pi^{2}\cdot E}$$
(20)b=0.50+0.65·a
(21)$$\omega =b+\sqrt{b^{2}-a}\;$$
where
 *λ_k_* Mechanical slenderness of the column. *E* Modulus of longitudinal elasticity of the material the column is made of. The Young’s modulus. *a, b* Nondimensional auxiliary coefficients. *ω* Nondimensional buckling coefficient greater than or equal to the unit, a function of the type of steel, yield stress, and mechanical slenderness of the member.

In the year 1976, the *Manual on Stability of Steel Structures* [27] published a series of buckling curves, where the ordinate axis showed the axial load related to the yield load and, in the abscissas axis, the slenderness ratio (Figure 2). These curves were obtained after the analysis of 1067 compression column tests performed in seven European countries (Belgium, France, Germany, Italy, Netherlands, United Kingdom, and Yugoslavia). The test program included various types of members (I,H,T, round, and square hollow sections) and it covered the range of slenderness ratios most frequently used in European constructional practice (55, 75, 95, 130, and 160). Later, a complementary program was carried out at Lehigh University on columns of relatively heavier sections, and the conclusions were used for establishing the buckling curves.

There was a total of three curves. Each one was valid for a certain type of section, and they considered whether the section was welded or rolled. The document gave parameters to transform those buckling curves to consider the dependence of their magnitude from the thickness of the compressed parts of the cross sections in buckling direction. Besides, the documents considered yield stress reduction factors based on geometrical characteristics of the cross sections as, for example, if the section had welded flange cover plates.

## 4. Operative Standards, Current Situation

In Spain, current buckling designs for steel columns follow the CTE DB SE A [27] and the EAE [28] standards, and in addition, the Structural Code [29] has recently been approved. This new Structural Code standard has a similar approach, in reference to the study of the buckling of steel bars, to the EC-3 and the repealed EAE. The two of them agree with both the ultimate limit state (ULS) and the Eurocode EC-3 [30], but the Spanish buckling coefficient “*ω*” has undergone a more sophisticated evolution as five buckling curves are advanced. The European buckling curves, proposed by the European Recommendation of Steel Construction [27] in 1976, appear to adapt better to the actual buckling phenomenon of steel columns.

These European buckling curves are slightly different to one another because they depend on the type of cross section, type of steel, steel plate thickness, and the axis, whether strong or weak, on which the member is likely to buckle. As a result, an elastic imperfection factor “*α*” is introduced for each of the cases.

Applying the buckling curves, a nondimensional buckling coefficient “*χ*” is obtained. This coefficient is roughly the inverse of the buckling coefficient “*ω*”, which, remotely but clearly, connects to the coefficients advanced by Rankine in his formulae.

Current European operative standards state that metallic members exhibiting simple cross section and centered compressive stress should verify Equation (22) so as not to undergo buckling.
(22)$$N_{b,Rd}=\chi \cdot A\cdot f_{yd}\;\geq N_{Ed}$$
where
 *N_b_,_Rd_* Design buckling resistance of a compression member. *N_Ed_* Design value of the compression force. *χ* Buckling coefficient less than or equal to the unit, a function of the imperfection factor and of the member’s reduced slenderness. *A* Gross cross-sectional area. *f_yd_* Design yield stress.

The nondimensional buckling coefficient “*χ*” can be obtained, applying
(23)$$\chi =\frac{1}{\phi +\sqrt{\phi^{2}-\lambda^{2}}}\;\leq 1$$
(24)$$\phi =0.50\cdot \left[1+\alpha \cdot \left(0.20-\lambda \right)+\lambda^{2}\right]\;$$
(25)$$\lambda =\sqrt{\frac{N_{pl}}{N_{crit}}}=\sqrt{\frac{A\cdot f_{y}}{\frac{\pi^{2}}{L_{k}^{2}}\cdot E\cdot I}}\;$$
where
 *χ* Nondimensional buckling coefficient less than or equal to the unit, a function of the member’s imperfection factor and reduced slenderness. *λ* Member’s reduced slenderness with respect to the selected axis. It should be less than or equal to 3. *α* Imperfection factor, dependent on the type of cross section and buckling axis. *I* Cross-sectional inertia dependent on the selected axis. *A* Gross cross-sectional area. *L_k_* Buckling length of the column. *f_y_* Yield stress of steel.

The ratio between reduced slenderness and mechanical slenderness of a member can readily be obtained when the following equation is applied:(26)$$\lambda =\lambda_{k}\cdot \sqrt{\frac{f_{y}}{\pi^{2}\cdot E}}=\frac{\lambda_{k}}{\lambda_{lim}}\;$$

The reduced slenderness “*λ*” is the square root of the coefficient “*a*” in Equation (19), and the coefficient “*ϕ*” matches roughly with the coefficient “*b*” in Equation (20). Therefore, the buckling coefficient “*χ*” is approximately the inverse of the buckling coefficient “*ω*” present in the Spanish standards MV and NBE for steel, already superseded, with the exception of the imperfection factor “*α*” in current operative standards, where five buckling curves, not the single one previously advanced, are proposed.

Figure 3 shows the ratio between compressive strength and mechanical slenderness for buckling steel columns proposed by current EC-3 [30], and by previous standards—already in disuse—together with Euler’s ideal hyperbola and Tetmajer’s experimental right line.

As can be observed, in the last 150 years, a long distance has been covered to obtain today’s buckling coefficients for slender metallic members likely to buckle. Practically, since Rankine’s proposals, the way has already been paved for advances to take place. Since then, every contribution made to the field has revolved around finding a gradually more sophisticated nondimensional coefficient that includes extra parameters without losing sight of the slenderness, the type of cross section, and the yield stress steel offers. These coefficients aim at reducing stress while confronting buckling, the yield stress, in steel members. The stress corrected and reduced multiplied by the cross section yields the maximum load the steel column can withstand, which, in turn, corrected by a series of safety coefficients—also evolving historically—achieves the purpose of obtaining the safe load that a column can withstand without buckling.

In Figure 4, the most representative historical standards using the same acceptable stress parameters of steel are grafted to compare column behavior when the strength is reduced while the slenderness is increased.

The type of steel selected is the current 275, exhibiting a yield stress of 275 N/mm^2^, ignoring any safety coefficient to avoid conflicts among the different standards.

Regarding American design codes, the American Institute of Steel Construction published specifications for structural steel buildings [10]. The main expression for buckling of nonslender elements under axial compression is
(27)$$F_{cr}=\left(0.658{\;}^{\frac{F_{y}}{F_{e}}}\right)\cdot F_{y}\;\mathrm{when}\;\frac{L_{C}}{r}\;\leq 4.71\;\sqrt{\frac{E}{F_{y}}}$$
(28)$$F_{cr}=0.877\cdot F_{e}\;\mathrm{when}\;\frac{L_{C}}{r}>4.71\;\sqrt{\frac{E}{F_{y}}}$$
where
 *E* Modulus of elasticity of steel. *F_e_* Elastic buckling stress through an elastic buckling analysis $F_{e}=\frac{\pi^{2}\cdot E}{{\left(\;\frac{L_{C}}{r}\right)}^{2}}$. *F_y_* Specified minimum yield stress of the type of steel being used. *r* Radius of gyration. *L_c_* Effective length of the member for buckling about the minor axis.

The Chinese “Code for Design of Steel Structures” [31] offers an expression for solid web beam columns, subjected to combined axial load and bending:(29)$$\frac{N}{\varphi_{x}A}+\frac{\beta_{mx}M_{x}}{{ɤ}_{x}W_{1x}\left(1-0.8\frac{N}{{N^{\prime }}_{Ex}}\right)}\leq f$$
where
 *N* Axial compression in the calculated portion of the member. *N’_Ex_* Parameter, *N’_Ex_* = *π^2^EA*/(1.1 *λ_x_^2^*). *φ_x_* Stability factor of axial compression members buckling in the plane of bending. *M_x_* Maximum moment in the calculated portion of the member. *W_1x_* Gross section modulus referred to the more compressed fiber in the plane of bending. *Β_mx_* Factor of equivalent moment, taken between 0.65–1.00. *f* Yield stress.

Let us finally highlight recent studies related to buckling, our main interest in this article. On the one hand, those works on elastic buckling on cold forming steel columns [32], those on buckling of plates [33], those on localized buckling on high-strength steel columns [34,35,36], and on the other, the studies of stress distribution by tension and by compression applied to steel [37], experimental and analytical studies on buckling [38,39,40,41], studies about buckling of restrained braces [42], and, finally, those on buckling behavior of rectangular plates [43].

## 5. Numerical Simulation

The objective of the numerical simulation was to verify the accuracy of the critical stress vs. mechanical slenderness curves shown in Figure 3 and Figure 4. For this, the commercial finite element software Abaqus [44] was employed. A total of 66 simulations were carried out on two different types of profiles widely used in construction.

### 5.1. Geometry

Two profile sections were studied: HEB-200 (universal column) and two-welded UPE-200 (two channel section) (Figure 5 and Table 1). HEB and UPE are two types of hot-rolled normalized European steel profiles. HEB profiles have H shape and UPE profiles have U shape. Every profile is followed by a number in the normalized European standard, which is the total height of the section. These two profiles are commonly used in construction to make steel columns and they are representative of open and closed sections. The length of the columns was modified from 250 to 30,000 mm, maintaining the cross section constant (33 lengths were studied for each column). Consequently, the mechanical slenderness $\lambda_{k}$ (Equation (5)) changed in each case from 2.5 up to a maximum value of 250 (33 slendernesses were studied for each column).

### 5.2. Finite Element Model Description

The employed finite elements are fully integrated, first-order, solid hexahedral elements with eight nodes and three degrees of freedom per node (C3D8). The size of each hexahedron was 1 cm, which offered accurate results with no sensitivity problems when the size slightly changed (Figure 6a). The boundary conditions are illustrated in Figure 6b. All the degrees of freedom of one end are fixed, and those of the other end are also fixed, except for the vertical displacement.

To achieve the initial geometrical imperfection that will produce buckling, an initial discrete load *F* was applied to the center of the profile and perpendicularly to its direction line (force *F* in Figure 6b). The magnitude of this force was chosen to produce a central bow of *L*/1000, *L* being the member length. This assumption safely covers unintentional load eccentricities at the member ends for the slenderness range of practical interest [28]. Therefore, the deviatoric load *F* varied depending on the slenderness of each specimen. After performing a sensibility analysis, it was concluded that, in all cases, this load was small enough to avoid significantly altering the critical buckling stress. Even when this initial load was applied to cause buckling along strong inertia axis, buckling along weak axis also took place first, as Figure 7 depicts. For this reason, only the buckling along the weak inertia axis was considered. Once the horizontal load was applied, vertical displacement *V* was imposed (Figure 6b). Displacement control was used to reach a vertical displacement *V* equal to 50 mm. Abaqus Standard static analysis [44] was employed, in which the solver is based on stiffness method and, consequently, gives solutions unconditionally stable, contrary to Abaqus Explicit analysis [44].

For the steel specimens, the elastoplastic model of Abaqus [44] was applied with the following parameter values: $E_{s}=210,000\;\mathrm{MPa},\;\nu =0.3,\;\mathrm{and}\;\sigma_{y}=275\;\mathrm{MPa},$ where $E_{s}$ is the elasticity modulus, ν is the Poisson coefficient, and $\sigma_{y}$ is the yield stress.

The finite element model was validated with experimental results on circular columns in previous research [45]. The same boundary conditions, applied loads, imposed displacements, material type, and element type were employed.

### 5.3. Results

Buckling took place in all specimens during the numerical simulations. Figure 8 shows the buckling shape of the HEB-200 and the two-welded UPE-200 for low and mid-slenderness ratios. The buckling stress was computed by performing the summation of the reaction forces on each node of the lower end and dividing the result by the profile area to obtain the equivalent compressive stress in column for all *V* displacement increments (Figure 6b) and, subsequently, finding the maximum equivalent compressive stress reached by the column [45].

The results of all numerical simulations are summarized in Figure 9 and Figure 10, where the critical buckling stress is represented according to the mechanical slenderness of the columns $\lambda_{k}$ (Equation (5)). Figure 9 contains the comparison between the finite element model and the different buckling theories (Euler, Tetmajer, Rankine, Secant, Barré, and Marvá) and Figure 10 compares the finite element model with the design standard curves (CTE, EC-3, AISC, MV-103, and NBE EA 95).

As can be seen in Figure 9, Euler’s model [7] overestimates the critical stress for low and middle mechanical slendernesses because it does not consider the nonlinear behavior of the steel when the yielding takes place. For high slenderness ratios, the onset of buckling occurs when no fiber of the cross section is yielded yet and, consequently, the Euler’s model [7] is more accurate. Nevertheless, as Figure 9b depicts, even for high slenderness ratios, Euler’s curve [7] is above the numerical model curves. This fact also occurs with the Tetmajer curve [13], whose formulation is the same as Euler’s one for high mechanical slenderness ratios ($\lambda_{k}\geq 105$). Below this mechanical slenderness value, the Tetmajer curve [12] is a straight line that overestimates the critical buckling stress at very low slendernesses (up to 30), underestimates it from $\lambda_{k}\geq 30$ to 90, and overestimates it again from $\lambda_{k}\geq 90$. Regarding Timoshenko’s secant curve [11], it underestimates the critical buckling stress up to $\lambda_{k}=210$, and beyond this slenderness, the value of the critical buckling stress almost matches the finite element model curves. Rankine’s curve [9] underestimates the critical buckling stress up to $\lambda_{k}=150$, from which this curve underestimates it. Along the range of mechanical slendernesses $30\leq \lambda_{k}\leq 80$, the underestimation of Rankine’s curve [9] is more accused. Finally, Barré–Marvá’s curve [9] underestimates the critical buckling stress for all mechanical slendernesses, mainly in the range $30\leq \lambda_{k}\leq 80$, as occurred with the Rankine’s curve [9]. As a global analysis, the order of the curves, considering their degree of overestimation of the critical buckling stress, is (1) Euler [7], (2) Tetmajer [12], (3) Rankine [9], (4) Timoshenko’s secant [11], (5) Barré–Marvá [8]. From a design point of view, Timoshenko’s secant [11] and Barré–Marvá [8] curves are on the safety side for any value of mechanical slenderness according to the finite element model results, but Timoshenko’s approximation [11] yields more optimized designs because it does not underestimate the critical buckling stress as much as Barré–Marvá.

Figure 10 compares the finite element model with the design standard curves. As can be seen, the AISC curve [10] overestimates the critical stress from slendernesses beyond 100 because its formulation is the same as Euler’s one [7] for high mechanical slenderness ratios. Nevertheless, it remains on the safe side with lower slendernesses. The rest of the curves (CTE-A, EAE and EC-3 [26,28,30], MV-103 [24], and NBE EA 95 [25]) generally underestimate the critical stress. This underestimation is more important when slendernesses range from 30 to 100. The curves of CTE-A, EAE, and EC-3 are divided into four types (“a0”, “a”, “b”, “c”, and “d”), depending basically on the profile type. The most conservative curve is CTE-A, EAE, and EC-3 “d” [26,28,30] for all slendernesses. The curves CTE-A, EAE, and EC-3 “a0” and “a” slightly overestimate the critical buckling stress for very high slendernesses ($\lambda_{k}\geq 160$ and $\lambda_{k}\geq 220$, respectively). The HEB-200 profile should be analyzed with the behavior of “c” curve according to CTE-A [26], EAE [28], and EC-3 [30], and the two-welded UPE-200 should be analyzed with the behavior of “c” curve according to CTE-A [26] and with “b” curve according EAE [28] and EC-3 [30]. This is in accordance with the numerical results, because the more accurate curve is “b” for two-welded UPE-200 and “c” for HEB-200 in comparison to the finite element model curves. For the two-welded UPE-200, the “b” curve is the more accurate with the numerical results without overestimating the critical stress at any slenderness. In case of HEB-200, for $\lambda_{k}\geq 200$, the critical stress of HEB profiles reduces with respect to that of two-welded UPE-200 and, consequently, approximates to the “b” curve. This fact causes that, at $\lambda_{k}=233$, the buckling curve of the two-welded UPE-200 surpasses “b” curve and, consequently, “b” curve overestimates the critical buckling length of two-welded UPE-200 profiles (Figure 10c).

## 6. Conclusions

Several accidents and collapses of structures have occurred due to buckling of steel structural elements, which have resulted in loss of a lot of human lives and very important material costs in the last 275 years.

Having considered the different arguments, the following ideas can safely be emphasized:

Buckling is a transcendent phenomenon for the safety of building and bridge structures. Buckling problems must be considered in the design, calculation, and construction phase of the structures.Every model is below Euler–Tetmajer’s equations, with the exception of that of AISC [10], proposed in 1921, which, for slenderness greater than 120, Euler’s equation [7] applies.Euler’s equation [7] has been applied in structural mechanics for almost 300 years with barely any modification. The fact that it is still being applied today makes it the most long-lived equation in structural engineering.Every model predicts that for mechanical slenderness of approximately 20, the member will not undergo the process of buckling.In practice, the differences that exist between standard MV 103 [24] (NBE EA 95 [25]) and standard CTE DB SE A, EAE, and EC-3 [26,28,30] are relatively small.The secant method formula yield values very much in line with those proposed by the standards.Rankine’s [9] and AISC [10] models yield values in line with those much more elaborated coming afterwards for mechanical slenderness of less than 100. Specifically, Rankine’s equation, even though old, is in tune with all the mechanical slenderness values predicted by much more sophisticated modern models.Older models, such as Barré–Marvá’s [8], yield more conservative values, somehow far from today’s actual values but quite acceptable nonetheless. The reason could well reside in the fact that those values were to be applied to cast iron columns that exhibited less slenderness than those used later.According to the performed numerical simulations, older models, such as Euler [7], Tetmajer [12], Rankine [9], and AISC [10], tend to slightly overestimate the critical buckling stress for high slenderness ratios from around 100 (except Euler, which considerably overestimates the critical stress for low and mid-slenderness ratios because it does not consider the nonlinear behavior of the steel when the yielding takes place). Besides, the Tetmajer [12] model also overestimates critical buckling stress for low slendernesses below approximately 30.The order of the theoretical curves, considering their degree of overestimation of the critical buckling stress, is (1) Euler [7], (2) Tetmajer [12], (3) Rankine [9], (4) Timoshenko’s secant [11], (5) Barré–Marvá [8]. From a design point of view, Timoshenko’s secant [11] and Barré–Marvá [8] curves are on the safe side for any value of mechanical slenderness according to the finite element model results.Design code models such as CTE-A, EAE and EC-3 models [26,28,30], MV-103 [24], and NBE EA 95 model [25], generally underestimate the critical buckling stress for any slenderness ratio according to the numerical simulations. CTE-A, EAE, and EC-3 models [26,28,30] show the most accurate results.

## Figures and Tables

**Figure 1 ijerph-18-12253-f001:**
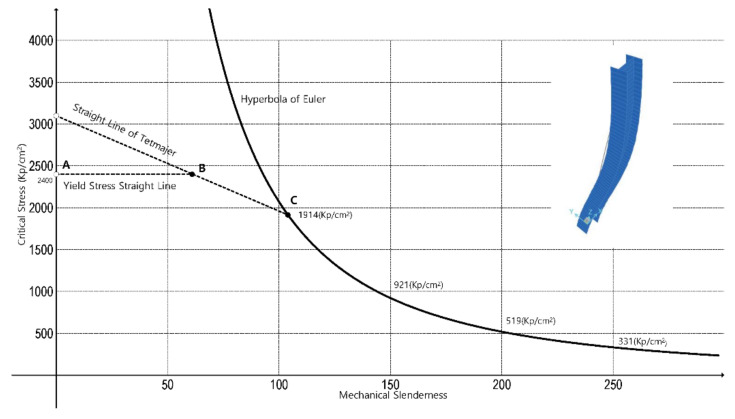
Basic diagram to show buckling of steel columns in the light of Euler’s equation, the yield stress of steel, and Tetmajer’s later equation.

**Figure 2 ijerph-18-12253-f002:**
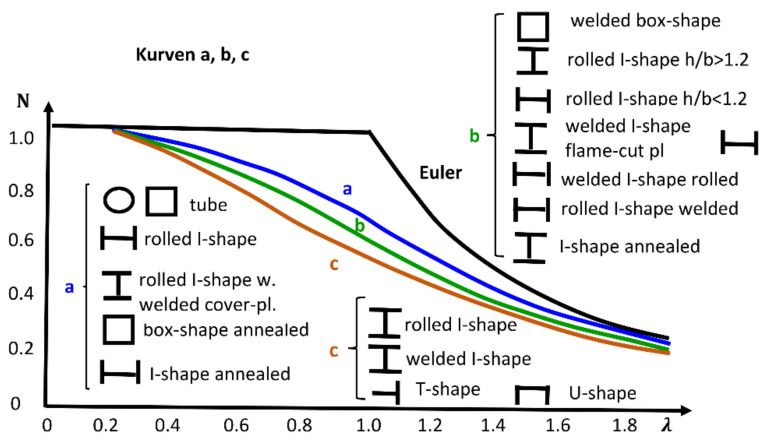
Buckling curves published in *Manual of Stability of Steel Structures* [27].

**Figure 3 ijerph-18-12253-f003:**
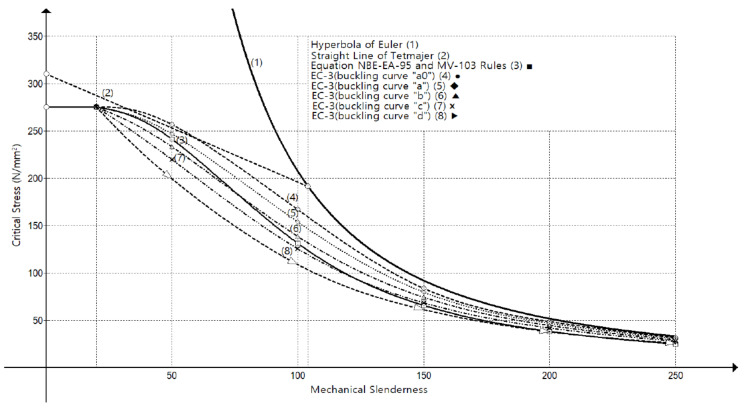
Strength–mechanical slenderness ratio diagram. Comparing different models from Spanish buckling standards for slender steel columns.

**Figure 4 ijerph-18-12253-f004:**
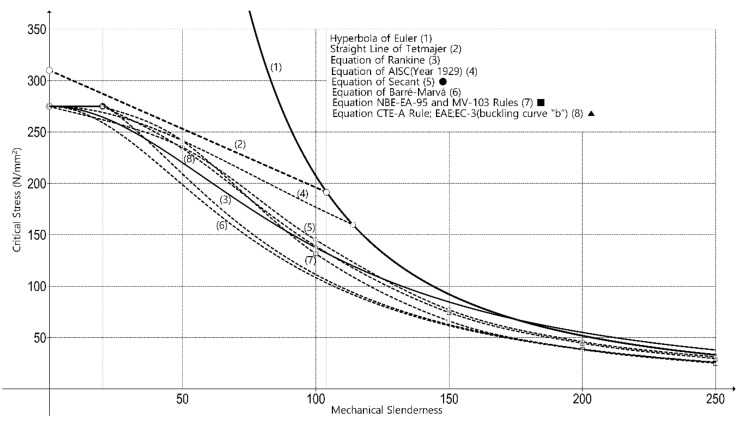
Strength–mechanical slenderness graph. Comparison between past and current models for slender steel column buckling.

**Figure 5 ijerph-18-12253-f005:**
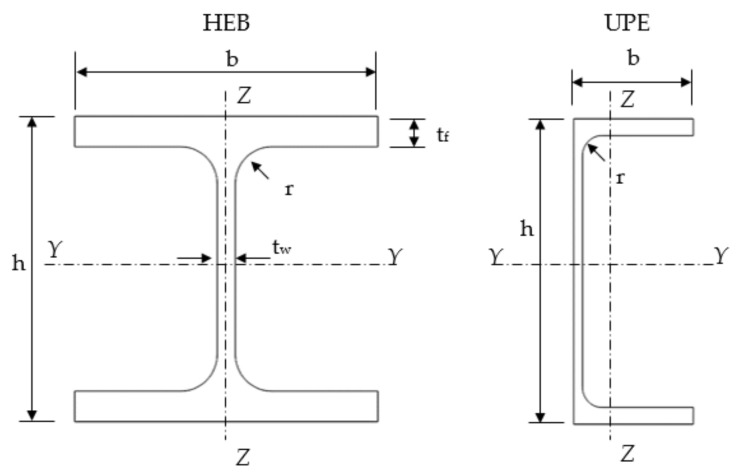
Mechanical characteristics of the profiles.

**Figure 6 ijerph-18-12253-f006:**
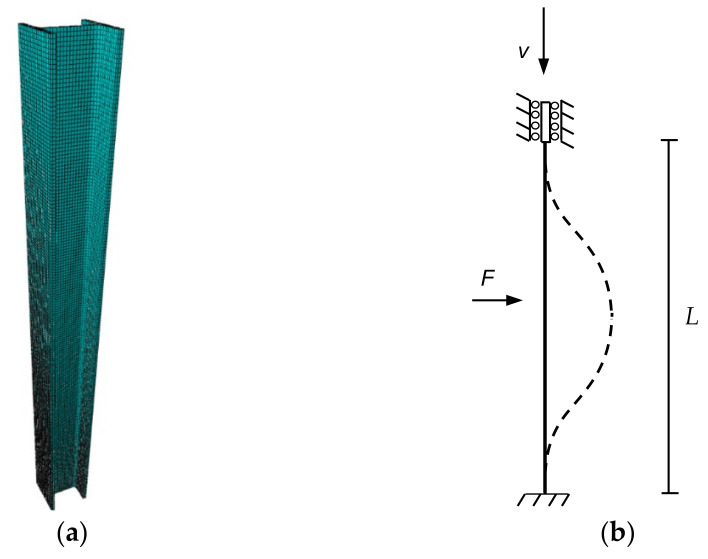
Specimen mesh and idealization: (**a**) Mesh for 5 m long specimen and mechanical slenderness of 49.4. (**b**) Boundary conditions, load, and displacement applied to the columns.

**Figure 7 ijerph-18-12253-f007:**
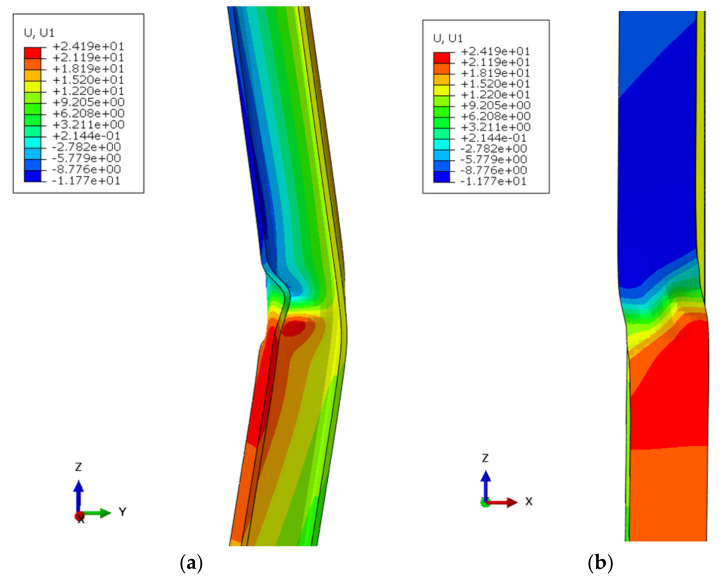
Displacements in X direction (mm) of buckled IPE-200 with initial load applied perpendicularly to the strong inertia axis, mechanical slenderness of 29.3: (**a**) front view; (**b**) lateral view.

**Figure 8 ijerph-18-12253-f008:**
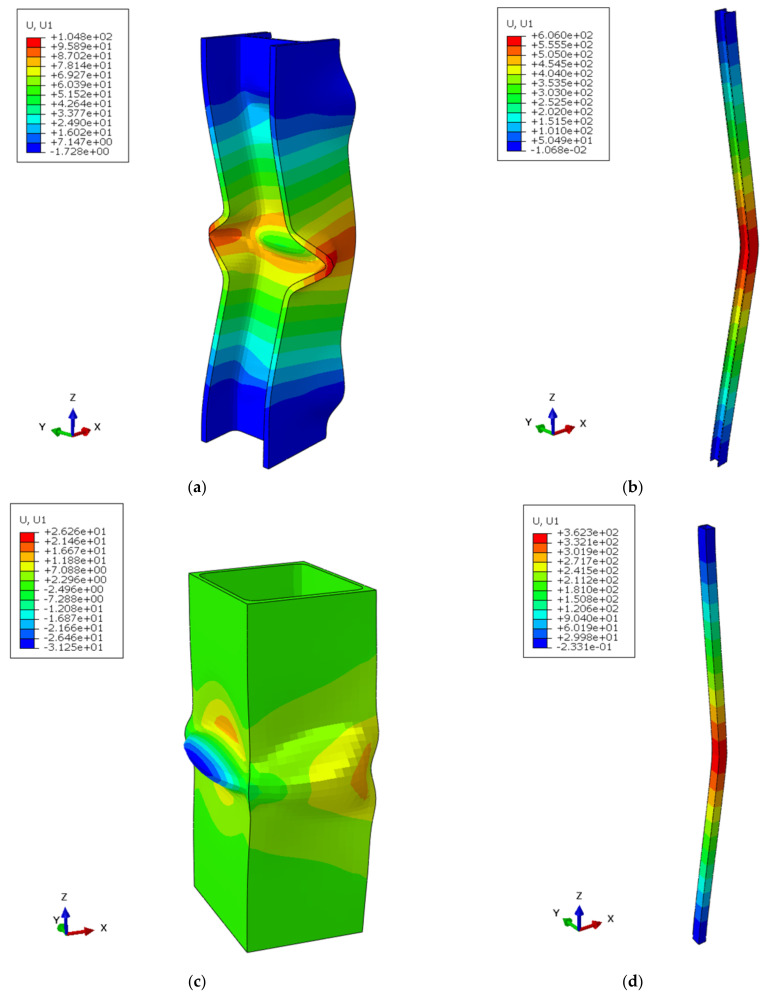
Displacements along X axis (mm) of buckling shapes: (**a**) HEB-200, mechanical slenderness of 9.9; (**b**) HEB-200, mechanical slenderness of 79; (**c**) two-welded UPE-200, mechanical slenderness of 4.2; (**d**) two-welded UPE-200, mechanical slenderness of 66.6.

**Figure 9 ijerph-18-12253-f009:**
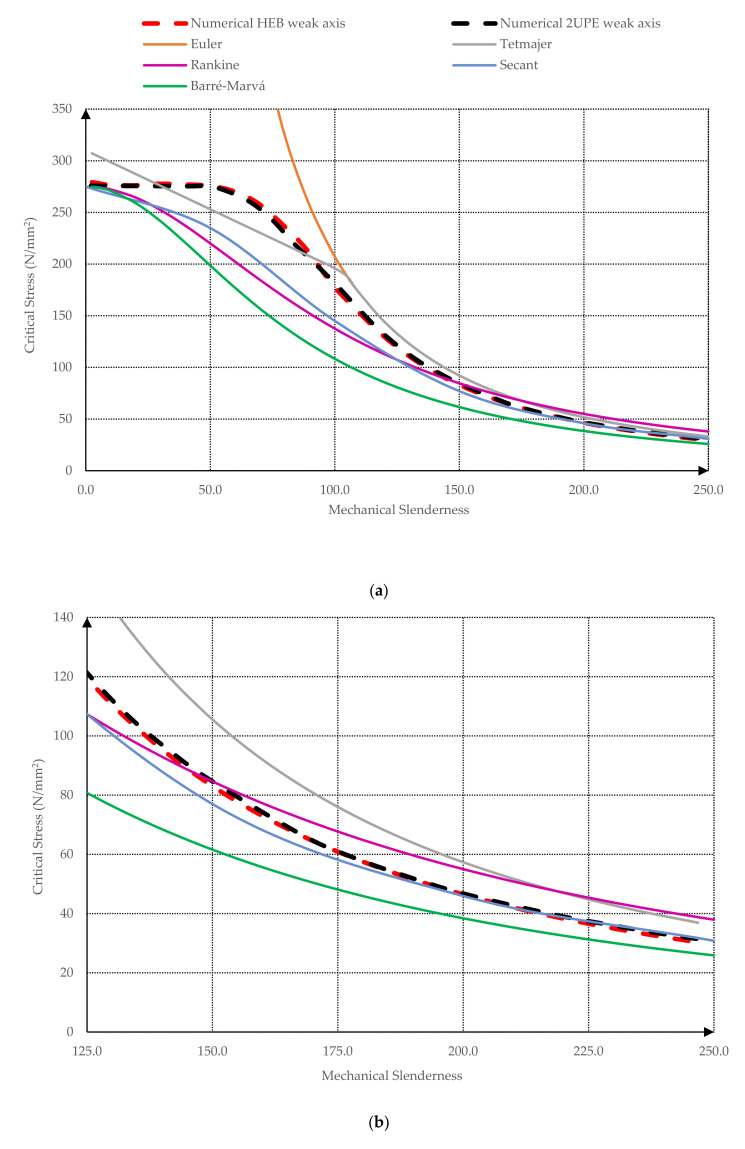
Strength–mechanical slenderness graph. Comparison theoretical models and numerical simulation: (**a**) mechanical slenderness from 0 to 250; (**b**) detail for high mechanical slenderness ratios (from 125 to 250).

**Figure 10 ijerph-18-12253-f010:**
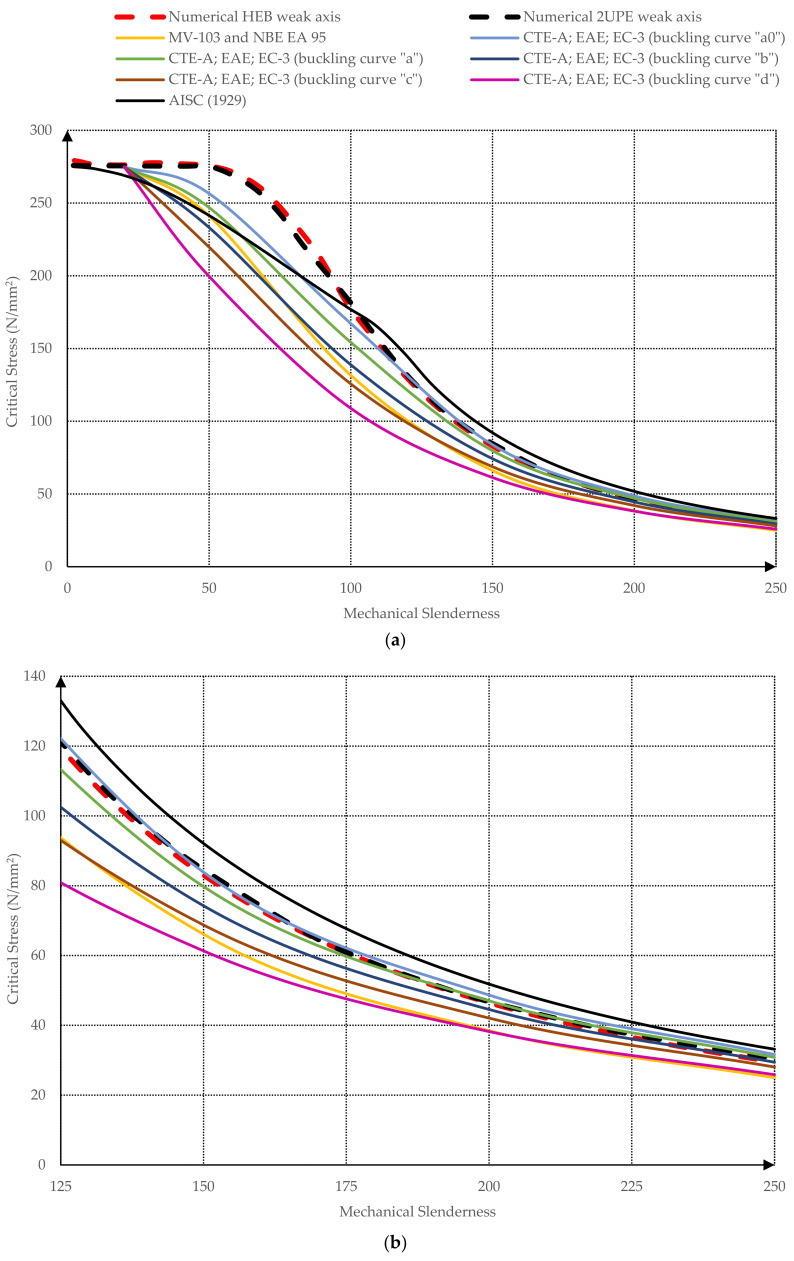
Strength–mechanical slenderness graph. Comparison design standard models and numerical simulation: (**a**) mechanical slenderness from 0 to 250; (**b**) detail for slenderness ratios from 125 to 250; (**c**) detail for slenderness ratios from 200 to 250.

**Table 1 ijerph-18-12253-t001:** Mechanical characteristics of the profiles.

Profile	h (mm)	b (mm)	r (mm)	t_w_ (mm)	t_f_ (mm)	Area (mm^2^)	Y Axis Inertia—I_y_ (mm^4^)	Z Axis Inertia—I_z_ (mm^4^)
HEB-200	200	200	18	9	15	78.1	5696	2003
UPE-200	200	80	13	6	11	29	1910	187
Two-welded UPE-200	200	160	13	9	11	58	3820	2090

## Data Availability

Not applicable.

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
