# Peer review of "Safety Issues in Buckling of Steel Structures by Improving Accuracy of Historical Methods"

_ijerph, 2021, doi:10.3390/ijerph182212253_

Round 1

Reviewer 1 Report

The authors present an interesting overall view of the bucking models. They have studied a comparative of the results obtained with these models with a numerical finite element model of compressed steel columns. The manuscript is correctly written, well-structured, well explained and very easy to read. Nevertheless, the following comments could be taken into consideration by the authors to improve the quality of the manuscript:

  1. Figure 1 shows the “Straight Line of Tetmajer” but this curve is not referenced with the same words in the text. Clarifying this would make it easier for readers to understand.
  2. About the current bucking design for steel columns in Spain, a new standard (Código Estructural) will be applicable from November 10th, 2021. Furthermore, the EAE standards will be repealed by the approval of this new standard. Even though the Código Estructural is not operative yet, the Authors could take into consideration to include some comments about this new operative standard in the manuscript. Otherwise, some of the comments in this study will be outdated very soon.   
  3. In Section 4.1 the Authors describe the two profile sections studied. To complete this information, could be interesting to indicate the cross-section of the profiles and its moment of inertia for the axis bucking considered. Could be interesting too if the Authors indicate the range of length of the profiles (minimum and maximum) considered in the analysis.
  4. In Figure 5 it is drawn the boundary conditions and the loads applied on the frame. This Reviewer suggests drawing on a dashed curve of the frame with its bucked shape.
  5. Figure 8 shows the same curves with different mechanical slenderness range and scales. Drawing a window of the zoomed area in Figure 8a might be helpful.
  6. This Reviewer suggests exchanging the order of the two first paragraphs of Section 5-Conclusions.

Author Response

Reviewer 1:

The authors present an interesting overall view of the bucking models. They have studied a comparative of the results obtained with these models with a numerical finite element model of compressed steel columns. The manuscript is correctly written, well-structured, well explained and very easy to read. Nevertheless, the following comments could be taken into consideration by the authors to improve the quality of the manuscript:

Authors thank the reviewer the time invested in reviewing this paper.

  1. Figure 1 shows the “Straight Line of Tetmajer” but this curve is not referenced with the same words in the text. Clarifying this would make it easier for readers to understand.

That's an interesting comment, thank you very much! The following information has been added to the manuscript:

When the members are characterized by a mechanical slenderness that falls between 0 and 105, the straight line of Tetmajer was proposed (Figure 1) by regression of his experimental data, which in stress units of kp and cm squared is the following:”

  1. About the current bucking design for steel columns in Spain, a new standard (Código Estructural) will be applicable from November 10th, 2021. Furthermore, the EAE standards will be repealed by the approval of this new standard. Even though the Código Estructural is not operative yet, the Authors could take into consideration to include some comments about this new operative standard in the manuscript. Otherwise, some of the comments in this study will be outdated very soon.

Thanks for the suggestion! A brief comment referring to the Structural Code has been introduced in the article to better define this section of the current regulations. The following information has been added to the manuscript:

“4. Operative standards, current situation

In Spain current buckling design for steel columns follow the CTE DB SE A [27] and the EAE [29] standards and in addition, the Structural Code [30] has recently been ap-proved. This new Structural Code standard has a similar approach, in reference to the study of the buckling of steel bars, to the EC-3 and the repealed EAE.

  1. In Section 4.1 the Authors describe the two profile sections studied. To complete this information, could be interesting to indicate the cross-section of the profiles and its moment of inertia for the axis bucking considered. Could be interesting too if the Authors indicate the range of length of the profiles (minimum and maximum) considered in the analysis.

HEB and UPE are two types of hot-rolled normalized European steel profiles. HEB profiles has H shape and UPE profiles has U shape. Every profile is followed by a number in the normalized European standard, which is the total heigh of the section.

Profile

h (mm)

b (mm)

r (mm)

tw (mm)

tf (mm)

Area (mm2)

Y axis inertia - Iy (mm4)

Z axis inertia - Iz (mm4)

HEB-200

200

200

18

9

15

78.1

5696

2003

UPE-200

200

80

13

6

11

29

1910

187

2-welded UPE-200

200

160

13

9

11

58

3820

2090

The following information has been added to the manuscript:

“Two profile sections were studied: HEB-200 (universal column) and two welded UPE-200 (two channel section) (Figure 5). HEB and UPE are two types of hot-rolled normalized European steel profiles. HEB profiles has H shape and UPE profiles has U shape. Every profile is followed by a number in the normalized European standard, which is the total heigh of the section. These two profiles are commonly used in construction to make steel columns and they are representative of open and closed sections. The length of the columns was modified from 250 to 30000 mm maintaining the cross-section constant (33 lengths were studied for each column). Consequently, the mechanical slenderness  (Equation (5)) changed in each case from approximately 2.5 up to a maximum value of 250.

Profile

h (mm)

b (mm)

r (mm)

tw (mm)

tf (mm)

Area (mm2)

Y axis inertia - Iy (mm4)

Z axis inertia - Iz (mm4)

HEB-200

200

200

18

9

15

78.1

5696

2003

UPE-200

200

80

13

6

11

29

1910

187

2-welded UPE-200

200

160

13

9

11

58

3820

2090

Figure 5. Mechanical characteristics of the profiles.

  1. In Figure 5 it is drawn the boundary conditions and the loads applied on the frame. This Reviewer suggests drawing on a dashed curve of the frame with its bucked shape.

Thank you so much for the suggestion, the Figure has been changed as follows:

  1. Figure 8 shows the same curves with different mechanical slenderness range and scales. Drawing a window of the zoomed area in Figure 8a might be helpful.

As reviewer 3 suggested, the Figure has been divided into two figures: The Figure 8 (now is Figure 9 because the Figure of the comment 3 has been added) has been divided into two figures. Figure 9 contains the comparison between the FEM and the different buckling theories (Euler, Tetmajer, Rankine, Secant, Barré and Marvá) and Figure 10 compares the FEM with the design standard curves (CTE, EC-3, AISC, MV-103 and NBE EA 95). Besides, new colors and a different type of graphic are employed. Figure 9.b and Figure 10.b and c are the zoom of the complete curves that are Figure 9.a and Figure 10.a. All these changes provide a clearer graphic and make easier the reading of the document.

  1. This Reviewer suggests exchanging the order of the two first paragraphs of Section 5-Conclusions.

Thank you for your comment! The suggestion has been considered and implemented.

Reviewer 2 Report

It is a good paper because it summarizes the historical background of bar buckling and the standard formulas of major countries.

1)P.10, Line 381   Please check font size, "Nb,Rd".

2)P.12, Line 440   "(mm)" is unnecessary.

3) Please check the chapter and section number.

        4.1 --> 5.1, 4.2--> 5.2, 4.3-->5.3, 5. Conclusions --> 6.Conclusions

4)From P.13 to P.14   In Sec.4.2, parameter "F, L, V" should be made italic.

Author Response

Reviewer 2:

It is a good paper because it summarizes the historical background of bar buckling and the standard formulas of major countries.

Authors thank the reviewer the time invested in reviewing this paper.

1)P.10, Line 381   Please check font size, "Nb,Rd".

Thank you for this comment! This question has been made.

2)P.12, Line 440   "(mm)" is unnecessary.

Thank you for this review! This suggestion has been considered.

3) Please check the chapter and section number.

        4.1 --> 5.1, 4.2--> 5.2, 4.3-->5.3, 5. Conclusions --> 6.Conclusions

Thank you for your suggestion! This improvement has been incorporated in the article.

4)From P.13 to P.14   In Sec.4.2, parameter "F, L, V" should be made italic.

Authors thank the reviewer for noticing this error. It has been fixed.

Reviewer 3 Report

It is nice to see a journal on public health address important engineering issues in a scientific and structured way.

This article reviews the historical theory and design of the buckling of steel columns. Overall, I found the paper well referenced and easy to read. There are a few minor mistakes that the authors should be able to catch with a thorough re-reading.

In the FEM images, I would suggest the authors adjust the clarity so the legend can be easily read.

My only real change or edit is related to Figure 8 and discussion associated. There are too many lines on Figure 8 to be read accurately. I would suggest that the authors separate the work into theory (Euler, Tetmajer, Rankine, Secant, Barr and Marva) and design (other curves) with the numerical curves present on both sets of graphs. This would be more meaningful and helpful to the audience.

In addition to separating the graphs, the discussion for Figure 8 could be expanded. The paper has a really short discussion section, which could be elaborated on by separating the graphs.

Author Response

Reviewer 3:

It is nice to see a journal on public health address important engineering issues in a scientific and structured way.

This article reviews the historical theory and design of the buckling of steel columns. Overall, I found the paper well referenced and easy to read. There are a few minor mistakes that the authors should be able to catch with a thorough re-reading.

Authors thank the reviewer the time invested in reviewing this paper.

In the FEM images, I would suggest the authors adjust the clarity so the legend can be easily read.

Thank you for the comment. The resolution of the FEM images has been improved. Now, the legend can be easily read.

My only real change or edit is related to Figure 8 and discussion associated. There are too many lines on Figure 8 to be read accurately. I would suggest that the authors separate the work into theory (Euler, Tetmajer, Rankine, Secant, Barr and Marva) and design (other curves) with the numerical curves present on both sets of graphs. This would be more meaningful and helpful to the audience.

Authors totally agree and thank so much the reviewer’s comment. The Figure 8 (now is Figure 9 because one more Figure has been added by suggestion of another reviewer) has been divided into two figures. Figure 9 contains the comparison between the FEM and the different buckling theories (Euler, Tetmajer, Rankine, Secant, Barré and Marvá) and Figure 10 compares the FEM with the design standard curves (CTE, EC-3, AISC, MV-103 and NBE EA 95). Besides, new colors are employed, a different type of graphic and more zoom has been implemented to make easier the reading of the document. All these changes provide a clearer graphic.

In addition to separating the graphs, the discussion for Figure 8 could be expanded. The paper has a really short discussion section, which could be elaborated on by separating the graphs.

Authors agree with the reviewer. The discussion has been extended as follows:

The results of all numerical simulations are summarized in Figure 9 and Figure 10, where the critical buckling stress is represented according to the mechanical slenderness of the columns  (Equation (5)). Figure 9 contains the comparison between the finite element model and the different buckling theories (Euler, Tetmajer, Rankine, Secant, Barré and Marvá) and Figure 10 compares the finite element model with the design standard curves (CTE, EC-3, AISC, MV-103 and NBE EA 95).

As can be seen in Figure 9, Euler’s model [7] overestimates the critical stress for low and middle mechanical slendernesses because it does not consider the non-linear behavior of the steel when the yielding takes place. For high slenderness ratios the onset of buckling occurs when no fiber of the cross section is yielded yet and, consequently, the Euler’s model [7] is more accurate. Nevertheless, as Figure 9 b) depicts, even for high slenderness ratios, Euler’s curve [7] is above the numerical model curves. This fact also occurs with the Tetmajer curve [13] whose formulation is the same of Euler’s one for high mechanical slenderness ratios (). Below this mechanical slenderness value, the Tetmajer curve [13] is a straight line that overestimates the critical buckling stress at very low slendernesses (up to 30), underestimates it from  to 90 and overestimates it again from . Regarding Timoshenko’s secant curve [12], it underestimates the critical buckling stress up to , and beyond this slenderness the value of the critical buckling stress almost matches the finite element model curves. Rankine’s curve [10] underestimates the critical buckling stress up to , from which this curve underestimates it. Along the range of mechanical slendernesses  the underestimation of Rankine’s curve [10] is more accused. Finally, Barré-Marvá’s curve [9] underestimates de critical buckling stress for all mechanical slendernesses, mainly in the range  as occurred with the Rankine’s curve [10]. As a global analysis, the order of the curves considering their degree of overestimation of the critical buckling stress is: (1) Euler [7], (2) Tetmajer [13], (3) Rankine [10], (4) Timoshenko’s secant [12], (5) Barré-Marvá [9]. From a design point of view, Timoshenko’s secant [12] and Barré-Marvá [9] curves are on the safety side for any value of mechanical slenderness according to the finite element model results, but Timoshenko’s approximation [12] yields more optimized designs because it does not underestimate the critical buckling stress as much as Barré-Marvá.

Figure 10 compares the finite element model with the design standard curves. As it can be seen, AISC curve [11] overestimates the critical stress from slendernesses beyond 100 because its formulation is the same of Euler’s one [7] for high mechanical slenderness ratios. Nevertheless it remains on the safety side with lower slendernesses. The rest of the curves (CTE-A, EAE and EC-3 [27,29,31], MV-103 [25] and NBE EA 95 [26]) generally underestimate the critical stress. This underestimation is more important when slendernesses ranging from 30 to 100. The curves of CTE-A, EAE and EC-3 are divided into four types (“a0”, “a”, “b”, “c” and “d”) depending basically on the profile type. The most conservative curve is CTE-A, EAE and EC-3 “d” [27,29,31] for all slendernesses. The curves CTE-A, EAE and EC-3 “a0” and “a” slightly overestimates the critical buckling stress for very high slendernesses ( and  respectively). The HEB-200 profile should be analyzed with the behavior of “c” curve according to CTE-A [27], EAE [29] and EC-3 [31] and the two welded UPE-200 should be analyzed with the behavior of “c” curve according to CTE-A [27] and with “b” curve according EAE [29] and EC-3 [31]. This is in accordance with the numerical results because the more accurate curve is “b” for two welded UPE-200 and “c” for HEB-200 in comparison to the finite element model curves. For the two welded UPE-200, the “b” curve is the more accurate with the numerical results without overestimating the critical stress at any slenderness. In case of HEB-200, for , the critical stress of HEB profiles reduces with respect to that of 2 welded UPE-200 and, consequently, approximates to the “b” curve. This fact causes that, at , the buckling curve of the two welded UPE-200 surpasses “b” curve and, consequently, “b” curve overestimates the critical buckling length of two welded UPE-200 profiles (Figure 10.c).

Besides, the following conclusions has been added:

  • The order of the theoretical curves considering their degree of overestimation of the critical buckling stress is: (1) Euler [7], (2) Tetmajer [13], (3) Rankine [10], (4) Timoshenko’s secant [12], (5) Barré-Marvá [9]. From a design point of view, Timoshenko’s secant [12] and Barré-Marvá [9] curves are on the safety side for any value of mechanical slenderness according to the finite element model results.
  • Design code models such us CTE-A, EAE and EC-3 models [27,29,31], MV-103 [25] and NBE EA 95 model [26], generally underestimate the critical buckling stress for any slenderness ratio according to the numerical simulations. CTE-A, EAE and EC-3 model [27,29,31] shows the most accurate results.

Reviewer 4 Report

Title: Safety issues in buckling of steel structures by improving accuracy of historical methods

Authors: Juan Carlos Pomares 1; Javier Pereiro-Barceló; Antonio González; Rafael Aguilar

General comment

In this study, the historical buckling models were compared with the finite element model for the steel columns subjected to compressive loads. From the comprehensive literature review, the history and existing models for the buckling were well investigate and the relationship between critical stress and mechanical slenderness was highlighted. By using the FEM-based critical stress-mechanical slenderness relationship, the existing buckling models were evaluated in terms of safety. The manuscript was well-written and easy to follow, but it is need to be discussed in more detail about the relationship between existing and FEM. Please refer to the comments below and resubmit the paper.

Technical comment

  1. (Line 446, Page 13). What is the abbreviation for HEB and UPE? Also, what does the number 200 after each abbreviation mean? I think it means a specific section that is usually used in the practical field, bur additional explanations of the two abbreviations are needed to help readers understand.
  2. (Line 474, Page 13). For the C3D8 element, it should be clearly mentioned in the manuscript that it is a “linear brick element” because the C3D8 element is not suitable for isochoric material behavior (i.e., plastic behavior).
  3. (Figure 5). Although the mesh size of 1 cm was mentioned in the manuscript, it will be helpful for readers to understand if the authors add the representation for the mesh of the column in Figure 5.
  4. The section number of “Numerical simulation” is 4, not 5. Please modify the section number.
  5. Figure 8 shows the overall relationship between all analyzed models and numerical simulation for strength-mechanical slenderness relationship, but the detailed comparison is not at a glance as described in the manuscript (Line 519-545). Therefore, it is recommended to draw a bar graph or a different graph that can be easily compared to the contents mentioned in the manuscript to help readers understand.

Author Response

Reviewer 4:

Title: Safety issues in buckling of steel structures by improving accuracy of historical methods

Authors: Juan Carlos Pomares 1; Javier Pereiro-Barceló; Antonio González; Rafael Aguilar

General comment

In this study, the historical buckling models were compared with the finite element model for the steel columns subjected to compressive loads. From the comprehensive literature review, the history and existing models for the buckling were well investigate and the relationship between critical stress and mechanical slenderness was highlighted. By using the FEM-based critical stress-mechanical slenderness relationship, the existing buckling models were evaluated in terms of safety. The manuscript was well-written and easy to follow, but it is need to be discussed in more detail about the relationship between existing and FEM. Please refer to the comments below and resubmit the paper.

Authors thank the reviewer the time invested in reviewing this paper.

Technical comment

  1. (Line 446, Page 13). What is the abbreviation for HEB and UPE? Also, what does the number 200 after each abbreviation mean? I think it means a specific section that is usually used in the practical field, bur additional explanations of the two abbreviations are needed to help readers understand.

HEB and UPE are two types of hot-rolled normalized European steel profiles. HEB profiles has H shape and UPE profiles has U shape. Every profile is followed by a number in the normalized European standard, which is the total heigh of the section.

Profile

h (mm)

b (mm)

r (mm)

tw (mm)

tf (mm)

Area (mm2)

Y axis inertia - Iy (mm4)

Z axis inertia - Iz (mm4)

HEB-200

200

200

18

9

15

78.1

5696

2003

UPE-200

200

80

13

6

11

29

1910

187

2-welded UPE-200

200

160

13

9

11

58

3820

2090

  1. (Line 474, Page 13). For the C3D8 element, it should be clearly mentioned in the manuscript that it is a “linear brick element” because the C3D8 element is not suitable for isochoric material behavior (i.e., plastic behavior).

Authors thank the interesting reviewer’s comment. According to the Abaqus documentation, the incompressible nature of plastic deformation in metals places limitations on the types of elements that can be used for an elastic-plastic simulation. The limitations arise because modeling incompressible material behavior adds kinematic constraints to an element; in this case the limitations constrain the volume at the element's integration points to remain constant. In certain classes of elements, the addition of these incompressibility constraints makes the element overconstrained. When these elements cannot resolve all of these constraints, they suffer from volumetric locking, which causes their response to be too stiff.

The fully integrated, second-order, solid elements available in Abaqus/Standard are very susceptible to volumetric locking when modeling incompressible material behavior and, therefore, should not be used in elastic-plastic simulations. The fully integrated, first-order, solid elements in Abaqus/Standard do not suffer from volumetric locking because Abaqus actually uses a constant volume strain in these elements. Thus, they can be used safely in plasticity problems. C3D8 is a fully integrated, first-order, solid element, therefore, it should be suitable to model plastic behavior, on the contrary to C3D20, which is a second-order element. C3D8R (eight-node brick element with reduced-integration) was not used by the researchers because more elements are needed (small elements are required to capture a stress concentration at the boundary of a structure) and these elements tend to be not stiff enough in bending.

The results of buckling analysis employing C3D8 elements was validated with experimental data in Pereiro-Barceló and Bonet (Pereiro-Barceló, J.; Bonet, J.L. Ni-Ti SMA bars behaviour under compression. Constr. Build. Mater. 2017, 155, 348–362, doi:10.1016/j.conbuildmat.2017.08.083.) and there are many authors in whose investigations use C3D8 to model plastic behavior.

Authors are aware that there are web pages such us (https://web.mit.edu/calculix_v2.7/CalculiX/ccx_2.7/doc/ccx/node26.html) where C3D8 is described and pointed out that it is not suitable for plastic behavior. However, the element they refer to is not the Abaqus one, but also another software (in case of the previous webpage, Calculix).

The following changes has been made in the manuscript:

“The employed finite elements are fully integrated, first-order, solid hexahedral elements with eight nodes and three degrees of freedom per node (C3D8). The size of each hexahedron was 1 cm, which offered accurate results with no sensitivity problems when the size slightly changed (Figure 6.a).”

  1. (Figure 5). Although the mesh size of 1 cm was mentioned in the manuscript, it will be helpful for readers to understand if the authors add the representation for the mesh of the column in Figure 5.

The figure 5 just represents an idealization of the structure to be able to see the boundary conditions and the applied load and displacement. A new figure (depicted below) has been added to show the representation of the mesh. A specimen of 5 m long has been chosen, which represents a mechanical slenderness of 49.4. This slenderness is low, but if a more slender specimen was chosen, the refined mesh would cause that the reader could not appreciate the elements in the mesh.

The following sub-figure has been added:

Figure 6. Specimen mesh and idealization: (a) Mesh for 5 m-long specimen and mechanical slenderness of 49.4 (b) Boundary conditions, load and displacement applied to the columns.

  1. The section number of “Numerical simulation” is 4, not 5. Please modify the section number.

Authors thank the reviewer for noticing this error. It has been fixed.

  1. Figure 8 shows the overall relationship between all analyzed models and numerical simulation for strength-mechanical slenderness relationship, but the detailed comparison is not at a glance as described in the manuscript (Line 519-545). Therefore, it is recommended to draw a bar graph or a different graph that can be easily compared to the contents mentioned in the manuscript to help readers understand.

Authors agree with the reviewer, the Figure must be clarified for the sake of understanding. As reviewer 3 suggested, the Figure has been divided into two figures: The Figure 8 (now is Figure 9 because one more figure has been added as a suggestion of a reviewer) has been divided into two figures. Figure 9 contains the comparison between the FEM and the different buckling theories (Euler, Tetmajer, Rankine, Secant, Barré and Marvá) and Figure 10 compares the FEM with the design standard curves (CTE, EC-3, AISC, MV-103 and NBE EA 95). Besides, new colors and a different type of graphic are employed. Figure 9.b and Figure 10.b and c are the zoom of the complete curves that are Figure 9.a and Figure 10.a. All these changes provide a clearer graphic and make easier the reading of the document.
